# KAN-Semi: A Semi-Supervised Approach Combining Self-Supervised Pre-training, Hierarchical Priors, and Kolmogorov-Arnold Networks for Landmark-based Biometry Estimation

## Abstract

Ultrasound (US)-based biometric estimation is crucial for monitoring labor progression and diagnosing fetal and maternal abnormalities. Reliable biometry estimation relies heavily on accurate landmark localization on standard planes, a process traditionally performed by sonographers. However, manual measurement is time-consuming, operator-dependent, and prone to variability. Although automated segmentation methods based on fully supervised models show promise, they often suffer from multi-stage error accumulation and a lack of expertly annotated data. To address these challenges, we introduce KAN-Semi, a semi-supervised network that combines self-supervised pre-training, hierarchical priors, and Kolmogorov-Arnold Networks (KANs). First, we utilize in-domain self-supervised pre-training with a Masked Autoencoder (MAE) to learn robust, domain-adapted representations for a novel CNN-ViT hybrid backbone. Next, we propose a Hierarchical Guidance Decoder, which encodes symbolic medical priors to regularize the model's reasoning, progressively guiding it from stable to variable structures. Finally, we explore Kolmogorov-Arnold Network (KAN)-enhanced heads as an alternative to conventional predictors, demonstrating their efficacy in complex spatial regression tasks. We perform extensive experiments on three intrapartum ultrasound datasets collected from 24 medical centers and institutions, showing that our approach significantly outperforms fully supervised models in landmark detection performance. Our work offers a structured framework for designing effective learning systems that integrate self-supervision, knowledge-based architectural design, and emerging network paradigms.

## 1 Introduction

Intrapartum ultrasound is a cornerstone of maternal and neonatal care, playing a critical role in safeguarding health during labor, a principle underscored by guidelines from bodies like the World Health Organization (WHO) (Organization et al., 2020) and the International Society of Ultrasound in Obstetrics and Gynecology (ISUOG) (Ghi et al., 2018). Despite its widespread use, its clinical impact remains constrained by significant challenges in accurately visualizing cranial landmarks and by substantial observer variability due to differences in operator skill (Youssef et al., 2017). Traditional manual assessment of fetal biometry, such as measuring the angle of progression (AoP), typically relies on single ultrasound images. The operator freezes a frame in a specific view and uses calipers to measure anatomical features. Some clinical guidelines even recommend repeating these measurements multiple times to ensure consistency. However, this manual process is not only time-consuming, particularly for less experienced practitioners, but also prone to both expected-value bias and selection bias.

Automating the process of landmark detection using artificial intelligence (AI) presents a promising solution to these challenges. AI can reduce variability, enhance measurement efficiency, and provide a more consistent, objective approach to biometry assessment. However, several technical barriers

need to be overcome for AI-based methods to be viable in real-world clinical practice. First, deep learning models typically require large annotated datasets for training, a well-documented challenge in the medical domain where expert annotations are labor-intensive and costly, while vast amounts of unlabeled data remain underutilized (Cheplygina et al., 2019). Second, some existing pipelines, such as those that first segment anatomical structures before identifying landmarks, can be prone to cumulative error propagation, limiting their efficiency and accuracy for real-time intrapartum use. Third, many landmark detection methods lack explicit anatomical priors, making accurate localization difficult when confronted with anatomical variability or incomplete views of the fetus. Finally, ultrasound images are frequently affected by artifacts such as speckle noise and acoustic shadowing, which further complicate automated landmark detection and necessitate the development of robust, generalizable models.

To systematically address these interconnected challenges of data scarcity, cumulative error, and anatomical variability, we propose **KAN-Semi**, a novel framework that integrates self-supervised pre-training, architectural priors, and an advanced semi-supervised fine-tuning pipeline. Our main contributions are threefold:

1. We leverage **in-domain self-supervised pre-training** with a Masked Autoencoder (MAE) (He et al., 2022) to learn robust, domain-adapted representations for a novel CNN-ViT hybrid backbone, effectively mitigating the effects of data scarcity and image artifacts.

2. We design a **Hierarchical Guidance Decoder** that explicitly encodes symbolic anatomical priors into the network architecture, guiding the model from stable to variable structures to handle anatomical variability and reduce localization ambiguity.

3. We conduct an early exploration of **Kolmogorov-Arnold Network (KAN) enhanced heads** as a substitute for conventional predictors, demonstrating their efficacy for precise spatial localization in a direct, end-to-end manner, thus avoiding the cumulative errors of multi-stage pipelines.

## 2 RELATED WORK

Our research is situated at the intersection of automated medical biometry, data-efficient learning, and advanced network architectures. This section reviews the most relevant prior work in these key domains to contextualize our contributions.

### 2.1 AUTOMATED LANDMARK DETECTION IN MEDICAL ULTRASOUND

The automation of biometric measurements in ultrasound is a long-standing goal. Early work by Youssef et al. (2017) on Angle of Progression (AoP) measurement established the feasibility of automated methods but also highlighted accuracy challenges compared to manual techniques. While deep learning has become the standard for related tasks, such as gestational sac segmentation (Danish et al., 2024), most fully supervised, multi-stage pipelines remain vulnerable to data scarcity and cumulative error propagation. The most proximate work, DSTCT by Jiang et al. (2024), successfully applies a semi-supervised model to the segmentation of the same anatomical structures. However, its architecture does not explicitly encode the hierarchical relationship between them. Our work differs by focusing on direct, end-to-end localization and introducing a novel architectural prior to leverage this anatomical knowledge.

### 2.2 DATA-EFFICIENT LEARNING IN MEDICAL IMAGING

To address the pervasive data scarcity in medical imaging, we leverage a two-stage data-efficient learning paradigm.

**Self-Supervised Pre-training** has emerged as a powerful technique to learn representations from unlabeled data. We focus on the generative approach of Masked Image Modeling (MIM), where the Masked Autoencoder (MAE) framework (He et al., 2022) stands as a state-of-the-art method, in contrast to contrastive methods like SimCLR (Chen et al., 2020). Critically, works like Models Genesis (Zhou et al., 2019) have demonstrated the superiority of *in-domain* pre-training over ImageNet pre-training for medical tasks, motivating our approach.

**Semi-Supervised Fine-tuning** further utilizes unlabeled data during the main training phase. The principle of consistency regularization, which evolved from early methods like the Π-Model (Rasmus et al., 2015) and Temporal Ensembling (Laine & Aila, 2016), is effectively implemented in the Mean Teacher framework (Tarvainen & Valpola, 2017) and has been refined in subsequent works like FixMatch (Sohn et al., 2020). Our KAN-Semi framework adopts this robust paradigm, which complements other data-efficient strategies like task-driven data augmentation (Chaitanya et al., 2019).

## 2.3 Advanced Network Architectures

Our model's architecture integrates several advanced design principles, moving beyond foundational keypoint detectors like DeepPose (Toshev & Szegedy, 2014) and Stacked Hourglass Networks (Newell et al., 2016).

**Hybrid CNN-Transformer Architectures** are now prominent, combining the spatial inductive biases of CNNs, built upon designs like ResNet (He et al., 2016) and EfficientNet (Tan & Le, 2019), with the global context modeling of Vision Transformers (ViTs) (Dosovitskiy et al., 2020). The synergy of this approach has been validated by works like CoAtNet (Dai et al., 2021) and UNETR (Hatamizadeh et al., 2022). Our work contributes a novel hybrid design while also noting that powerful pure-CNN (e.g., ConvNeXt (Liu et al., 2022)) and pure-Transformer (e.g., Swin-Unet (Cao et al., 2022)) backbones are typically knowledge-agnostic.

**Kolmogorov-Arnold Networks (KANs)**, recently proposed by Liu et al. (2024), represent a fundamental shift from traditional MLP design by using learnable splines as activation functions on network edges. Their application to dense prediction tasks like heatmap regression is still in its infancy, and our work contributes some of the first empirical evidence in this emerging area.

## 3 Methodology

We propose **KAN-Semi**, a comprehensive two-stage learning pipeline designed to address the challenges of automated landmark detection in ultrasound. Our framework first learns powerful domain-specific representations via self-supervision, then fine-tunes a knowledge-informed, semi-supervised model for the localization task. The overall architecture is illustrated in Figure 1.

### 3.1 Stage 1: Self-Supervised Pre-training for Representation Learning

A significant challenge in medical imaging is the domain gap between general-purpose datasets like ImageNet and the specialized characteristics of ultrasound imagery. To overcome this, our paradigm begins with a dedicated self-supervised pre-training stage on all available in-domain ultrasound images. We adapt the Masked Autoencoder (MAE) framework (He et al., 2022) for this purpose.

The MAE operates on the principle of masked image modeling: a high percentage of image patches are randomly masked, and a ViT-based encoder is trained on the remaining visible patches to produce a latent representation from which a lightweight decoder reconstructs the original masked content. The learning objective is to minimize the Mean Squared Error (MSE) between the reconstructed and original patches, computed only over the masked set $\mathcal{M}$:

$$\mathcal{L}_{\text{MAE}} = \frac{1}{|\mathcal{M}|} \sum_{i \in \mathcal{M}} \|\mathbf{x}_i - \hat{\mathbf{x}}_i\|^2 \tag{1}$$

where $\mathbf{x}_i$ is the $i$-th patch from the original image and $\hat{\mathbf{x}}_i$ is its reconstruction by the decoder. This demanding task forces the encoder to learn robust, high-level semantic representations. After pre-training, we retain the ViT encoder's weights to serve as a powerful, domain-adapted initialization for our downstream model.

### 3.2 Stage 2: Semi-Supervised Fine-tuning of KAN-Semi

In the second stage, the pre-trained encoder is integrated into our main KAN-Semi model, which is then fine-tuned using a semi-supervised approach. The student model within this framework is composed of three key architectural components.

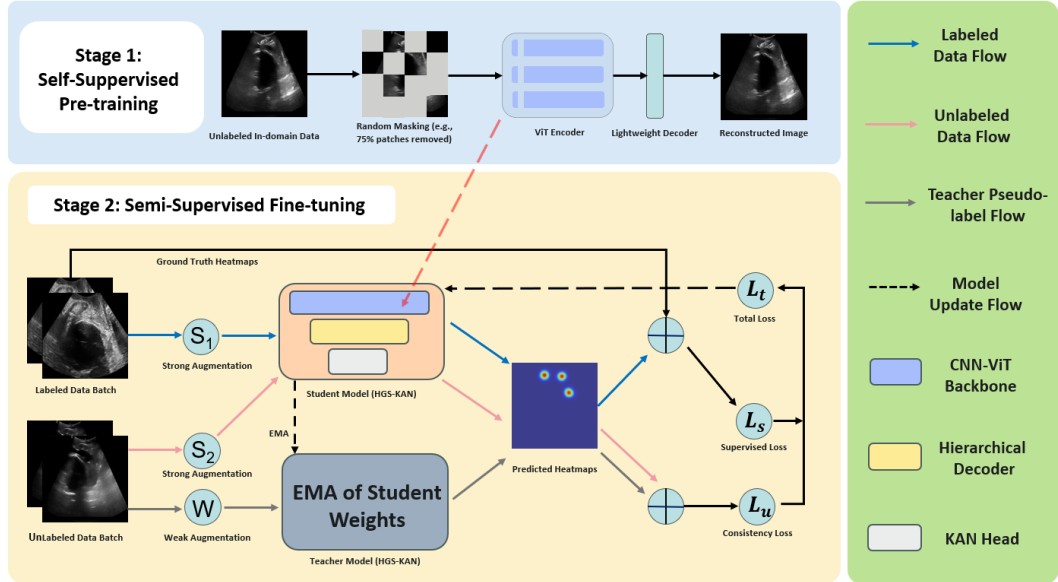

Figure 1: The overall architecture of our proposed KAN-Semi framework, which consists of two main stages. **Stage 1 (Self-Supervised Pre-training):** A Vision Transformer (ViT) encoder is pre-trained on all available in-domain unlabeled data using a Masked Autoencoder (MAE) objective. The learned weights provide a powerful, domain-adapted initialization. **Stage 2 (Semi-Supervised Fine-tuning):** The pre-trained ViT is integrated into the bottleneck of our HGS-KAN student model. The student model is then fine-tuned using both labeled and unlabeled data within a Mean Teacher framework. The total loss ($\mathcal{L}_t$) combines a supervised loss ($\mathcal{L}_{sup}$) on labeled data and a consistency loss ($\mathcal{L}_{unsup}$) on unlabeled data.

### 3.2.1 HYBRID CNN-TRANSFORMER BACKBONE

Our network backbone is a hybrid architecture based on the successful U-Net architecture (Ronneberger et al., 2015), which is prized for its effective use of skip connections. We employ an EfficientNet-B4 (Tan & Le, 2019) as the primary CNN encoder for its parameter efficiency and strong feature extraction. At the U-Net's bottleneck—the point of highest semantic abstraction—we insert the lightweight ViT module pre-trained in Stage 1. This ViT bottleneck acts as a global context aggregator, modeling long-range dependencies between anatomical structures from the high-level feature maps provided by the CNN encoder.

### 3.2.2 HIERARCHICAL GUIDANCE DECODER

To explicitly incorporate anatomical priors, we introduce the Hierarchical Guidance Decoder, which mimics an "easy-to-hard" expert reasoning process. As detailed in Figure 2, let $\mathcal{F} \in \mathbb{R}^{H \times W \times C}$ be the shared feature map from the backbone. The process is formalized as:

$$\boldsymbol{H}_{base} = h_{base}(\mathcal{F}) \tag{2}$$

$$\mathcal{F}_{guided} = \text{Concat}\left(\mathcal{F}, \text{sg}(\boldsymbol{H}_{base})\right) \tag{3}$$

$$\boldsymbol{H}_{adv} = h_{adv}(\mathcal{F}_{guided}) \tag{4}$$

where $h_{base}$ and $h_{adv}$ are the base and advanced prediction heads. The base heatmaps ($\boldsymbol{H}_{base}$) for stable landmarks are predicted first. They are then concatenated with $\mathcal{F}$—after a stop-gradient (sg) operation—to form a guided feature map, $\mathcal{F}_{guided}$. This map is then used to predict the advanced heatmaps ($\boldsymbol{H}_{adv}$) for the more variable landmarks.

### 3.2.3 KAN-ENHANCED PREDICTION HEAD

We challenge the standard use of a simple $1 \times 1$ convolution for the final feature-to-heatmap mapping by proposing a KAN-Enhanced Prediction Head (KANHead), based on Kolmogorov-Arnold

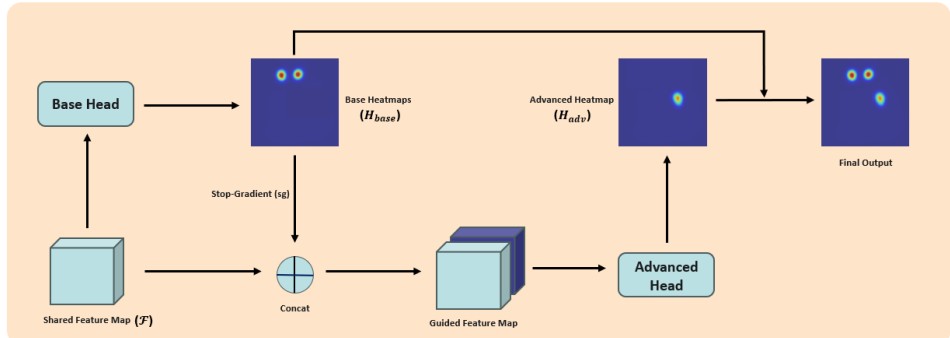

Figure 2: The architecture of our Hierarchical Guidance Decoder. The shared feature map ($\mathcal{F}$) from the backbone is first used by a Base Head to predict the base heatmaps for stable landmarks ($\boldsymbol{H}_{base}$). These heatmaps, after a stop-gradient (sg) operation, are concatenated with the original feature map to form a guided feature map. An Advanced Head then uses this guided map to predict the heatmap for the more variable landmark ($\boldsymbol{H}_{adv}$). Finally, the base and advanced heatmaps are concatenated along the channel dimension to form the complete multi-channel output.

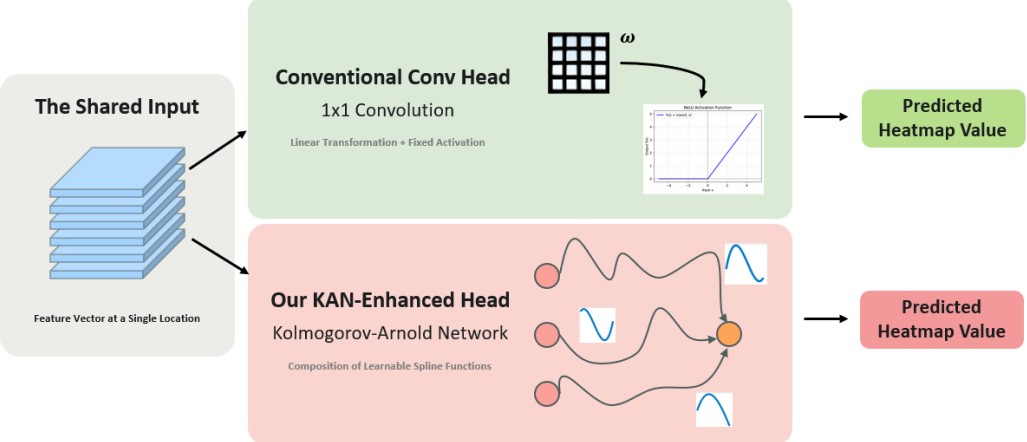

Figure 3: Conceptual comparison between a conventional head and our proposed KAN-Enhanced Head. **Left (Conventional):** A standard $1 \times 1$ convolution applies a linear transformation followed by a fixed activation (e.g., ReLU). **Right (Our KAN-Head):** A KAN composes learnable, univariate spline functions on its edges, allowing for a more expressive mapping.

Networks (KANs) (Liu et al., 2024). As illustrated in Figure 3, unlike a conventional head which uses a linear transformation with a fixed activation, a KAN composes learnable spline activation functions on its edges. This provides superior non-linear modeling capabilities, allowing for a more expressive and efficient mapping from features to heatmap intensities. Our work provides early empirical validation of KANs for this complex, dense regression task.

## 3.3 REGULARIZATION STRATEGIES FOR ROBUSTNESS

To enhance generalization, we employ two regularization strategies: **Spatial Regularization** via Dropout2d in the decoder, and **Label Space Regularization**, where we add Gaussian noise to the ground-truth keypoint coordinates before generating heatmaps to act as a form of label smoothing for regression.

### 3.4 Learning Objective and Semi-Supervised Strategy

The fine-tuning stage is driven by a composite loss within the Mean Teacher (Tarvainen & Valpola, 2017) semi-supervised framework. The total loss for the student model is:

$$\mathcal{L}_{\text{total}} = \mathcal{L}_{\text{s}} + \lambda(t) \cdot \mathcal{L}_{\text{u}} \tag{5}$$

The supervised loss, $\mathcal{L}_{sup}$, is the Mean Squared Error (MSE) between the student's predictions on a strongly-augmented labeled batch and the perturbed ground-truth heatmaps. The consistency loss, $\mathcal{L}_{unsup}$, is the MSE between the student's predictions on a strongly-augmented unlabeled batch and the pseudo-labels generated by the teacher model. The student's parameters, $\theta_s$, are updated via gradient descent on $\mathcal{L}_{\text{total}}$. The teacher's parameters, $\theta_t$, are an exponential moving average (EMA) of the student's parameters:

$$\theta_t \leftarrow \beta\theta_t + (1 - \beta)\theta_s \tag{6}$$

where $\beta$ is the EMA decay rate. The consistency weight $\lambda(t)$ and the decay rate $\beta$ are dynamically scheduled during training to stabilize the learning process.

## 4 Experiments

We conduct a series of experiments to evaluate KAN-Semi. We aim to answer: (1) How does our architecture compare to strong baselines? (2) What is the contribution of each component? (3) How effective is our semi-supervised strategy?

### 4.1 Experimental Setup

#### 4.1.1 Dataset and Metrics

Our study is validated on a large-scale, multi-center dataset from **24 medical centers**, collected under IRB approval. The raw collection contains 53,996 frames from 434 videos, all of which are used for our MAE self-supervised pre-training. For the downstream fine-tuning task, we use a curated subset of expertly annotated standard-plane frames, which is split into a training set (2,431 labeled, 5,497 unlabeled), a validation set (443 labeled), and a test set (501 labeled). We evaluate performance using two primary metrics: **Mean Radial Error (MRE)** in pixels, measuring localization precision, and **Absolute Progression Difference (APD)** in degrees, calculated as mean($|\text{AoP}_{\text{pred}} - \text{AoP}_{\text{gt}}|$) to assess clinical measurement accuracy.

#### 4.1.2 Implementation Details

Our entire framework was implemented using PyTorch and trained on NVIDIA RTX 4090 GPUs with 24GB of memory. Key hyperparameters are summarized in Table 1.

**Stage 1 (MAE Pre-training):** We pre-train a ViT-Tiny encoder for 200 epochs on $224 \times 224$ images with a 75% masking ratio, using the AdamW optimizer.

**Stage 2 (Fine-tuning):** The KAN-Semi model, featuring an EfficientNet-B4 backbone and our MAE pre-trained ViT bottleneck, is fine-tuned for 200 epochs on $512 \times 512$ images. We employ extensive data augmentations and two key regularization strategies: Dropout2d (p=0.1) and label perturbation ($\sigma = 2.0$ pixels). The model is trained with a batch size of 24 (12 labeled, 12 unlabeled) using AdamW and a cosine annealing learning rate scheduler with a 5-epoch warmup. Our dynamic Mean Teacher strategy involves a 30-epoch ramp-up for the consistency weight and EMA decay, and a confidence threshold ramping from 0.5 to 0.9.

### 4.2 Main Performance Comparison

We first compare the architectural merits of KAN-Semi against several strong and representative baselines under a fair, fully-supervised setting. We then present the result of our full semi-supervised model to demonstrate the additional gains from leveraging unlabeled data. The results are presented in Table 2.

The results in Table 2 clearly establish the superior performance of our proposed approach. First, when focusing on the fully-supervised setting to compare architectural merits, our KAN-Semi model

Table 1: Key hyperparameters for our two-stage training process.

| Stage 1: MAE Pre-training | | Stage 2: Fine-tuning | |
|---|---|---|---|
| Parameter | Value | Parameter | Value |
| Encoder | ViT-Tiny | CNN Encoder | EfficientNet-B4 |
| Image Size | $224 \times 224$ | Image Size | $512 \times 512$ |
| Masking Ratio | 0.75 | Optimizer | AdamW |
| Epochs | 200 | Initial LR | $5 \times 10^{-4}$ |
| Batch Size | 64 | LR Scheduler | Cosine Annealing |
| Optimizer | AdamW | Warmup Epochs | 5 |
| Learning Rate (LR) | $1.5 \times 10^{-4}$ | Total Epochs | 200 |
| Weight Decay | 0.05 | Labeled Batch Size | 12 |
| | | Unlabeled Batch Size | 12 |
| | | Dropout Probability | 0.1 |
| | | Label Perturb. $\sigma$ | 2.0 |
| | | EMA Decay | $0.99 \rightarrow 0.999$ |
| | | Confidence Thresh. | $0.5 \rightarrow 0.9$ |

Table 2: Quantitative comparison with representative baseline methods on the test set. All methods are trained in a fully-supervised setting, except for our final model. Best results are in **bold**.

| Method | Training Setting | MRE (pixels) ↓ | APD (°) ↓ |
|---|---|---|---|
| *Classic and Hybrid Baselines* | | | |
| U-Net (Ronneberger et al., 2015) | Fully-Supervised | 16.48 | 6.98 |
| TransUNet* (Chen et al., 2021) | Fully-Supervised | 17.60 | 7.01 |
| *Modern Representative Backbones* | | | |
| Swin-Unet* (Cao et al., 2022) | Fully-Supervised | 16.51 | 6.89 |
| ConvNeXt-Unet* (Liu et al., 2022) | Fully-Supervised | 16.05 | 5.77 |
| *Our Proposed Framework* | | | |
| KAN-Semi (ours) | Fully-Supervised | **15.54** | **5.74** |
| KAN-Semi (ours) | Semi-Supervised | **14.45** | **4.99** |

achieves the best performance with an MRE of 15.54 pixels. It not only surpasses the classic U-Net and TransUNet but also outperforms models equipped with powerful modern backbones like Swin-Unet and ConvNeXt-Unet. Notably, while the ConvNeXt-Unet achieves a strong APD of 5.77°, our model matches this performance while achieving a lower MRE, indicating more precise landmark localization overall. This highlights the intrinsic advantages of our design, where the synergy of a domain-adapted ViT backbone, hierarchical guidance, and a KAN-enhanced head is more effective than relying on a powerful general-purpose backbone alone.

Second, the impact of our full two-stage paradigm is demonstrated by comparing the semi-supervised version of KAN-Semi to all other methods. By leveraging unlabeled data, our model achieves a new state-of-the-art with an MRE of 14.45 pixels and, most critically, a significantly lower APD of 4.99°. The substantial improvement from its own fully-supervised version (a 7.0% relative reduction in MRE and a 13.5% relative reduction in APD) provides clear validation for our semi-supervised strategy. This comprehensive comparison confirms that our framework, which combines a superior architecture with an effective data-efficient learning strategy, is a more robust and accurate solution for this challenging task. A detailed breakdown of each component's contribution will be presented in the following ablation studies.

### 4.3 IN-DEPTH ABLATION STUDIES AND ANALYSIS

To understand the individual contribution and synergistic effects of our core components, we conduct a comprehensive ablation study. We start from our full KAN-Semi framework and systematically deactivate or replace key innovations. All models in this study were trained under the same semi-supervised setting for a fair comparison. The results are detailed in Table 3.

Table 3: Ablation study on the contributions of our core components. All models are trained under the semi-supervised setting. HG denotes Hierarchical Guidance.

| # | Configuration | MAE+ViT | HG | KAN | MRE (pix)↓ | APD (°)↓ |
|---|---|---|---|---|---|---|
| 1 | KAN-Semi (ours) | ✓ | ✓ | ✓ | **14.45** | **4.99** |
| *Ablating Architectural Components* | | | | | | |
| 2 | w/o Hierarchical Guidance | ✓ | ✗ | ✓ | 14.53 | 5.18 |
| 3 | w/o KAN Head | ✓ | ✓ | ✗ | 14.70 | 5.81 |
| 4 | w/o HG and KAN | ✓ | ✗ | ✗ | 14.89 | 5.30 |
| *Ablating Pre-training Strategy* | | | | | | |
| 5 | w/o MAE+ViT (CNN-only) | ✗ | ✓ | ✓ | 15.73 | 6.72 |

### 4.3.1 ANALYSIS OF ARCHITECTURAL COMPONENTS

The results in Table 3 highlight the importance of our two primary architectural innovations. Removing the Hierarchical Guidance (HG) decoder (Row 2 vs. Row 1) leads to a notable performance drop, which confirms the value of our knowledge-informed design. By decomposing the problem into an "easy-to-hard" sequence, the architecture is effectively regularized, improving localization robustness.

The impact of the KAN-Enhanced Head is even more pronounced. Replacing the KAN Head with a conventional predictor (Row 3 vs. Row 1) results in a substantial degradation, especially in the clinical APD metric (from 4.99° to 5.81°). To understand why, we visualize a representative spline activation function learned by our KAN-Head in Figure 4. Unlike the fixed, piece-wise linear ReLU function, our KAN head learns a smooth, highly non-linear mapping. This demonstrates its superior expressive power to capture the complex relationships between features and heatmap values, which is critical for high-precision localization.

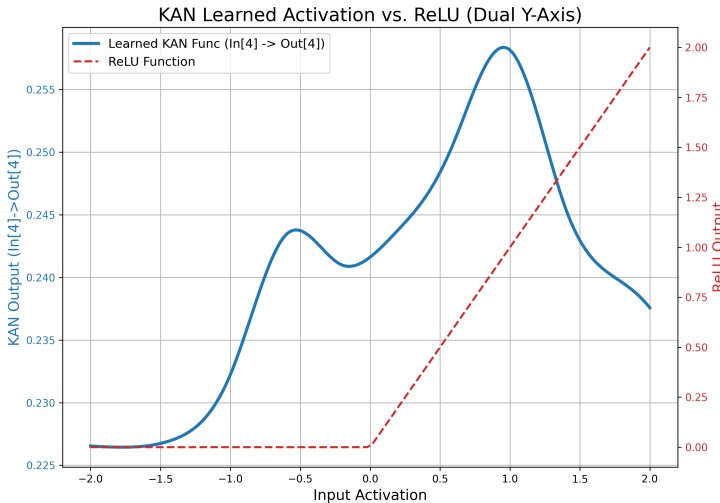

Figure 4: Visualization of a representative spline activation function learned within our KAN-Enhanced Head, compared to the fixed ReLU function. The learned function exhibits a smooth, highly non-linear behavior, demonstrating its superior expressive capability over standard fixed activations.

### 4.3.2 THE CRITICAL ROLE OF SELF-SUPERVISED REPRESENTATION

The most significant performance drop is observed when ablating our two-stage learning paradigm. By replacing the MAE pre-trained ViT backbone with a standard CNN-only backbone (Row 5 vs. Row 1), the MRE deteriorates sharply by 1.28 pixels to 15.73, and the APD worsens to 6.72°. This result provides unequivocal evidence that our in-domain, self-supervised pre-training strategy is the

most critical factor for success, building a powerful and domain-adapted feature foundation upon which our architectural innovations can thrive. In summary, the ablation study confirms that all three components are essential and synergistic contributors to the overall success of the KAN-Semi framework.

### 4.4 QUALITATIVE RESULTS

To provide a more intuitive understanding of our model's robustness, Figure 5 presents a visual comparison of landmark detection results on two particularly challenging cases from our test set. We compare our full KAN-Semi model against our strongest fully-supervised baseline, ConvNeXt-Unet.

The top row showcases a case with severe acoustic shadowing that obscures a significant portion of the fetal head. The baseline model is visibly distracted by this artifact, erroneously placing the fetal head landmark far from the ground truth. In contrast, our KAN-Semi model, likely benefiting from the robust representations learned during in-domain MAE pre-training, successfully ignores the artifact and provides a precise localization.

The bottom row presents a case with low overall contrast and indistinct tissue boundaries around the pubic symphysis. The baseline model struggles with this ambiguity, resulting in noticeable errors for all three landmarks. Our model, however, demonstrates superior resilience to the poor image quality, with its predictions closely aligning with the ground-truth annotations.

These qualitative examples visually corroborate the quantitative results from our main experiments. They highlight our framework's enhanced robustness in clinically realistic scenarios, demonstrating its ability to overcome common challenges like image artifacts and low signal-to-noise ratios where strong baseline models may fail.

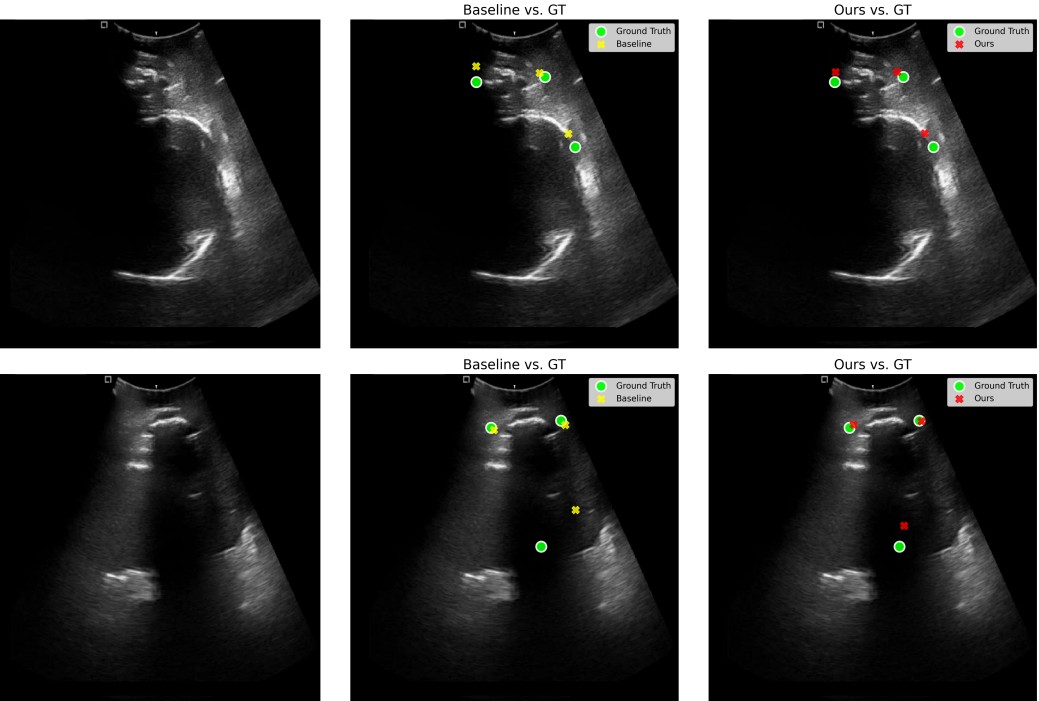

Figure 5: Qualitative comparison on challenging cases from the test set. For each case (row), we show the original image, the predictions of our strongest baseline (ConvNeXt-Unet, ✗), and the predictions of our KAN-Semi model (✗), all overlaid with the ground truth (●).

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

# A   APPENDIX

## A.1   THE USE OF LARGE LANGUAGE MODELS (LLMS)

In the preparation of this manuscript and the accompanying source code, we utilized a large language model (LLM), specifically a version of Google's Gemini Pro, as a general-purpose assistive tool. In accordance with ICLR 2026 policy, the usage of the LLM was confined to the following aspects.

**1. Writing Assistance and Polishing.** The LLM was employed to aid in the drafting and refinement of the manuscript's text. This included tasks such as improving sentence structure, ensuring grammatical correctness, rephrasing for clarity, and checking for consistency in style across different sections. For example, initial drafts of technical descriptions were collaboratively refined with the LLM to enhance their readability and formal academic expression.

**2. Literature Retrieval and Discovery.** The LLM served as an advanced search tool to help identify relevant prior work and contextualize our contributions. For instance, it assisted in finding foundational papers for concepts like Masked Autoencoders and Kolmogorov-Arnold Networks, and

helped identify recent, high-performance baseline models for comparison. This process acted as a supplement to our own comprehensive literature review using traditional academic search engines.

**3. Code Refinement and Debugging.** The LLM was also used to assist with code development. Its role included helping to refactor certain code blocks for better readability and efficiency, suggesting alternative implementations for specific functions, and assisting in debugging by identifying potential errors or suggesting troubleshooting steps.

It is important to emphasize that all core scientific contributions—including the initial research ideation, the design of the KAN-Semi framework's core logic, the implementation of the overall experimental pipeline, the execution and final analysis of experiments, and the conclusions drawn— were conducted entirely by the human authors. The LLM's role was strictly that of an assistive tool for writing, information retrieval, and code refinement. The authors take full responsibility for all content presented in this paper, including the correctness of the source code, the accuracy of the technical claims, and the validity of the experimental results.

### A.2   CODE AND REPRODUCIBILITY

The complete source code for our KAN-Semi framework is provided as supplementary material to facilitate the verification of our results and to ensure reproducibility. The implementation includes both core stages of our methodology: the MAE-based self-supervised pre-training and the semi-supervised fine-tuning. Detailed instructions for environment setup, data preparation, and execution of the training pipeline are available in the accompanying `README.md` file.

