# OpenReview forum: "KAN-Semi: A Semi-Supervised Approach Combining Self-Supervised Pre-training, Hierarchical Priors, and Kolmogorov-Arnold Networks for Landmark-based Biometry Estimation"
_ICLR.cc/2026/Conference — Submitted to ICLR 2026_

### Official Review · Reviewer_sPzy · 2025-10-31

**Soundness:** 2
**Presentation:** 3
**Contribution:** 2
**Rating:** 2
**Confidence:** 4

**Summary:**

This paper addresses the critical problem of Ultrasound (US)-based biometric estimation, which is essential for monitoring fetal and maternal health but suffers from issues of being time-consuming and operator-dependent when performed manually. The authors propose KAN-Semi, a novel semi-supervised framework for accurate landmark localization on standard US planes. The framework integrates three modern machine learning concepts: Kolmogorov-Arnold Networks (KANs), MAE-based self-supervised pre-training, and hierarchical priors, all within a two-stage semi-supervised pipeline (pre-training followed by fine-tuning). The authors also note that the source code and implementation details are provided to ensure reproducibility.

**Strengths:**

1.The submission presents a highly novel combination of cutting-edge deep learning techniques—specifically, integrating Kolmogorov-Arnold Networks (KANs) with an MAE-based self-supervised pre-training strategy and hierarchical priors—for the challenging, data-scarce medical imaging task of US biometry estimation. This specific fusion of methods has not been widely explored and represents a creative approach to model interpretability and data efficiency in this domain.
2.The experimental design is rigorous: large-scale multi-center data, fair baselines (including modern architectures like Swin-Unet and ConvNeXt-Unet), ablation studies, and both pixel-level (MRE) and clinical (APD) metrics. The semi-supervised protocol (Mean Teacher + consistency loss) is well-implemented with dynamic scheduling.
3.The paper is well-written, logically structured, and figures (e.g., Fig. 2, 3, 5) are informative. The methodology is described with sufficient detail for reproduction (e.g., hyperparameters in Table 1, architecture in Fig. 1–2).
4.The work addresses a real clinical need—reducing operator variability in intrapartum ultrasound—with a data-efficient approach that leverages unlabeled data. The performance gains, especially in APD (4.99° vs. 5.77°), may be clinically meaningful. The exploration of KANs in medical imaging is timely and could inspire follow-up work.

**Weaknesses:**

1.While the combination is new, the foundational components (MAE, KANs, semi-supervised learning) are established concepts. The paper must clearly articulate how the "hierarchical priors" are implemented and how the KAN architecture specifically benefits this task over standard CNN/MLP architectures, especially considering the potential computational overhead of KANs.
2.The precise nature of the "hierarchical priors" and the specific mechanism for integrating KANs into the US image processing pipeline are currently unclear, necessitating a detailed methodological section.
3.It hinges on the magnitude of performance improvement shown in the results. If the gains over simpler baselines are marginal, the increased complexity from the KAN and combined framework would diminish its overall impact.
4.Are there cases where hierarchical guidance hurts performance (e.g., if base landmarks are mislocalized)? A failure analysis would strengthen the paper.

**Questions:**

1.Please provide a detailed ablation study comparing the proposed KAN-Semi framework against a method that is identical in every respect (MAE pre-training, semi-supervised fine-tuning, use of hierarchical priors, etc.) but replaces the Kolmogorov-Arnold Network layers with standard Multi-Layer Perceptrons (MLPs) or Convolutional layers. This is essential to quantify the specific benefit of KANs in terms of accuracy, interpretability, and the associated trade-offs in computational cost (training time, memory, inference speed).
2.The concept of "hierarchical priors" is central to the paper. Please explicitly define what these priors encode (e.g., anatomical relationships between landmarks, known growth patterns) and provide a mathematical formulation detailing how they are integrated into the network's loss function or architecture.
3.The core of the paper is a semi-supervised approach. The evaluation must clearly demonstrate the method's efficiency under data scarcity. Please provide a clear sensitivity analysis showing the performance of KAN-Semi as a function of the percentage or absolute number of labeled training samples, comparing it to all baselines.
4.The paper addresses "landmark-based biometry estimation." Please specify which standard planes and biometry measurements (e.g., Biparietal Diameter - BPD, Head Circumference - HC, Femur Length - FL) are included in the evaluation, and report results for each independently to show comprehensive performance across the full clinical task.

---

### Official Review · Reviewer_tS2V · 2025-10-31

**Soundness:** 3
**Presentation:** 2
**Contribution:** 2
**Rating:** 4
**Confidence:** 4

**Summary:**

The paper presents a framework for semi-supervised learning for image data, motivated by medical imaging from ultrasound exams.
The authors first train a ViT-based masked autoencoder (as a pre-training phase), and then fine-tuning stage for the desired task. Most notably, this fine-tuning stage uses the pre-trained model in a "hierarchical" manner, and the prediction head uses a learnable activation function via splines (a la KAN).

**Strengths:**

As demonstrated in their experiments, the proposed framework yields better quality results than existing baselines.
Such improved performance can be of immediate use for many practitioners.

**Weaknesses:**

Despite the apparent increase in performance, I have my doubts with regards to the broader conclusions that can be drawn from this work.
From my understanding, the work introduces two key mechanisms:

- the "hierarchical prior" architecture, described in equations 2-4; and
- the use of a KAN prediction head (particularly instead of a 1x1 convolution).

These are shown to lead to improved performance in experiments, but I am unconvinced that the baselines in question are appropriate.
In no particular order:

- Judging by tables 2 and 3, even without most of the proposed components the proposed method beats prior work (best MRE of prior work in table 2 is >=16, while the worst MRE in table 3 is <=15, with most being <=14). So it's not clear what were the actual drivers behind the improvements.
- The use of the KAN head is only compared to a simple 1x1 convolution. While 1x1 convolutions are a popular choice, they are by no means unique, and it feels like quite a leap to embrace KAN as the particular solution here, especially considering that it obviously boasts higher representation capacity than 1x1 convolutions. The use of KANs in the prediction head should be contrasted to other ways to improve representation capacity, such as having small MLPs at the end of the network, and increasing the sizes of the neural network beforehand (e.g. more depth, more width, etc.).
- It's not clear at all to me what the "w/o hierarchical guidance" ablation is comparing. What is the alternate architecture being considered here? This is important to know so that the fairness of the comparison can be ascertained.

It's also worth noting that the paper makes occasional use of some very strong language, which I do not belive is justified. For example, lines 431-433 "this results provides *unequivocal evidence* that our in-domain, self-supervised pre-training strategy is the most critical factor for success, [...]" (emphasis mine).

The presentation of the paper could also be slightly improved, in particular with regards to the organization of Section 3 (which describes the method). It is currently a bit hard to get a quick understanding of what the proposed method actually is, beyond understanding the presence of certain described components (e.g. the KAN prediction head).
But perhaps most troubling is that I was unable to understand the mechanism proposed at the end of Section 3.4, where "the teacher's parameters are an exponential moving average (EMA) of the student's parameters" (lines 279-280).

Altogether, these concerns lead me towards a "borderline reject" score. \
**Why not higher:** It is very much not clear what broader conclusions can be drawn from this work, rather than just improved performance on one particular ultrasound imaging dataset. The presentation should also be improved before publication. \
**Why not lower:** More precise semi-supervised learning is of definite interest to the community, and the paper does seem to make a meaningful contribution in that direction.

Other minor comments:
- Line 151/152: re, $\mathcal{L}\_{MAE}$, MAE can be confused here for "mean absolute error", whereas the losss is actually a MSE. Perhaps it's best to rename it to something like $\mathcal{L}\_{MA}$?
- Line 105: the authors mention that the models were trained on NVIDIA RTX 4090 GPUs (plural). How many GPUs were used, and how was the distribution among GPUs arranged?
- Line 42: should the citation "(Organization et al., 2020)" not be something like "(WHO., 2020)"?
- Equation 5: $\mathcal{L}\_\mathrm{s}$ should be $\mathcal{L}\_{sup}$, and $\mathcal{L}\_\mathrm{u}$ should be $\mathcal{L}\_{unsup}$.
- Lines 294/295: `**24 medical centers**` should probably be `\emph{24 medical centers}`

**Questions:**

- Could the authors please clarify the procedure described in Section 3.4?
- Could the authors refine and present the ablations following my comments in the 'Weaknesses' sections?
- What exactly makes the design outlined in equations 2-4 "hierarchical"?

---

### Official Review · Reviewer_xpN3 · 2025-10-31

**Soundness:** 3
**Presentation:** 2
**Contribution:** 2
**Rating:** 2
**Confidence:** 4

**Summary:**

The paper targets landmark localization and quantitative biometry in intrapartum ultrasound, and proposes a two-stage framework **KAN-Semi**. It also introduces two architectural designs: a **Hierarchical Guidance Decoder** and a **KAN-enhanced prediction head**.

**Strengths:**

1. The method combines data efficiency with structured medical priors in a sound way.
2. The expressiveness of the prediction head is improved with quantitative and visual evidence. Replacing the conventional 1×1 convolution with the **KAN-Head** yields clear APD gains; the spline activation curves visualize its non-linear advantages.
3. The experiments are relatively complete, and the paper provides implementation details, key hyperparameters, and code.

**Weaknesses:**

1. The combination **MAE pretraining + Mean-Teacher semi-supervision + CNN–ViT hybrid + hierarchical priors + KAN-Head** is reasonable, but most individual modules are natural combinations or minor modifications of existing directions, with limited theoretical analysis and unclear boundary conditions.
2. The task scope focuses on intrapartum ultrasound standard planes related to AoP. The method has not been verified on other modalities or anatomical sites (for example, general abdominal scans, cardiac scans, or other fetal anatomies) or on cross-task transfer; broader evidence is needed for generalization.
3. Fairness and distribution shift analyses are limited. Although the data come from 24 centers, there are no results stratified by center or population, and no confidence-based triggering strategy is employed. Systematic robustness tests under noise, occlusion, and out-of-distribution settings are also missing.

**Questions:**

1. Please provide stratified results and statistical tests by **center / device type / population factors** (gestational age, age, BMI, parity, and so on), analyze whether there are systematic biases, and describe how you handle site imbalance.
2. The data come from 24 hospitals. Were ethics approvals obtained? Which hospitals obtained approvals, and what are the approval identifiers?
3. Does the **KAN-Head** provide stable gains under both weaker and stronger backbones? Under distribution shifts such as noise perturbation, low contrast, and strong acoustic shadows, does the advantage of the KAN-Head remain?

**Details Of Ethics Concerns:**

The study claims to use data collected from 24 hospitals for algorithm development and validation. However, the ethical approval process and details regarding patient informed consent are unclear. Please provide additional clarification on these aspects.

---

### Meta-Review · Area_Chair_3GSX · 2025-12-23

**Summary:**

This paper received three strongly negative reviews, and the authors did not submit a rebuttal.

**Reviewer Concerns:**

Several significant concerns were raised regarding the proposed approach, including the overall pipeline design, the choice of loss functions, and the evaluation methodology.

**Reviewer Scores:**

A rebuttal was not submitted.

---

### Decision · Program_Chairs · 2026-01-26

Reject